# FETA: Towards Specializing Foundation Models for Expert Task Applications

**Amit Alfassy\*[1,3]   Assaf Arbelle\*[1]   Oshri Halimi[1,3]   Sivan Harary[1]**
**Roei Herzig[1]   Eli Schwartz[1]   Rameswar Panda[1]   Michele Dolfi[1]   Christoph Auer[1]**
**Kate Saenko[2,4]   Peter W. J. Staar[1]   Rogerio Feris[2]   Leonid Karlinsky\*[2]**

[1]IBM Research, [2]MIT-IBM AI-Watson Lab, [3]Technion, [4]Boston University

## Abstract

Foundation Models (FMs) have demonstrated unprecedented capabilities including zero-shot learning, high fidelity data synthesis, and out of domain generalization. However, as we show in this paper, FMs still have poor out-of-the-box performance on expert tasks (e.g. retrieval of car manuals technical illustrations from language queries), data for which is either unseen or belonging to a long-tail part of the data distribution of the huge datasets used for FM pre-training. This underlines the necessity to explicitly evaluate and finetune FMs on such expert tasks, arguably ones that appear the most in practical real-world applications. In this paper, we propose a first of its kind FETA benchmark built around the task of teaching FMs to understand technical documentation, via learning to match their graphical illustrations to corresponding language descriptions. Our FETA benchmark focuses on text-to-image and image-to-text retrieval in public car manuals and sales catalogue brochures. FETA is equipped with a procedure for completely automatic annotation extractaion, allowing easy extension of FETA to more documentation types and application domains in the future. Our automatic annotation leads to an automated performance metric shown to be consistent with metrics computed on human-curated annotations (also released). We provide multiple baselines and analysis of popular FMs on FETA leading to several interesting findings that we believe would be very valuable to the FM community, paving the way towards real-world application of FMs for practical expert tasks currently "overlooked" by standard benchmarks focusing on common objects.

## 1   Introduction

Foundation Models (FMs) is a broad term, relating to models that through their training on huge data acquire "skills" going beyond the base training objectives [6]. They are commonly trained using hundreds of millions of data points and a collection of base tasks either uni-modal, e.g. only language, or multi-modal, e.g. text-image pairs. Remarkably, the skills acquired by FMs demonstrate very good transferability to a wide variety of new downstream tasks, many times with very limited or no data for the target task. Since their introduction in the Natural Language Processing (NLP) domain [6, 12, 46], FMs have been applied to uni-modal [8, 12, 46] and multi-modal [1, 14, 21, 30, 31, 32, 45, 52, 60, 62, 63] Vision & Language (V&L) scenarios, as well as demonstrated unprecedented capabilities for high fidelity data synthesis [40, 47, 49] and out of domain generalization [48]. However, despite the tremendous progress in FMs many gaps still remain open with regards to reaching human level performance in some mundane tasks [56], as well as in many human expert ones. In particular, for many types of 'specialized' data (e.g. illustrated technical, scientific documentation, medical, or other expert domains data), which are of the utmost interest for many real-world applications, FM performance is still lacking in many respects due to: (i) specialized

36th Conference on Neural Information Processing Systems (NeurIPS 2022) Track on Datasets and Benchmarks.

Common Objects Vision & Language Task Data        FETA Expert Task data

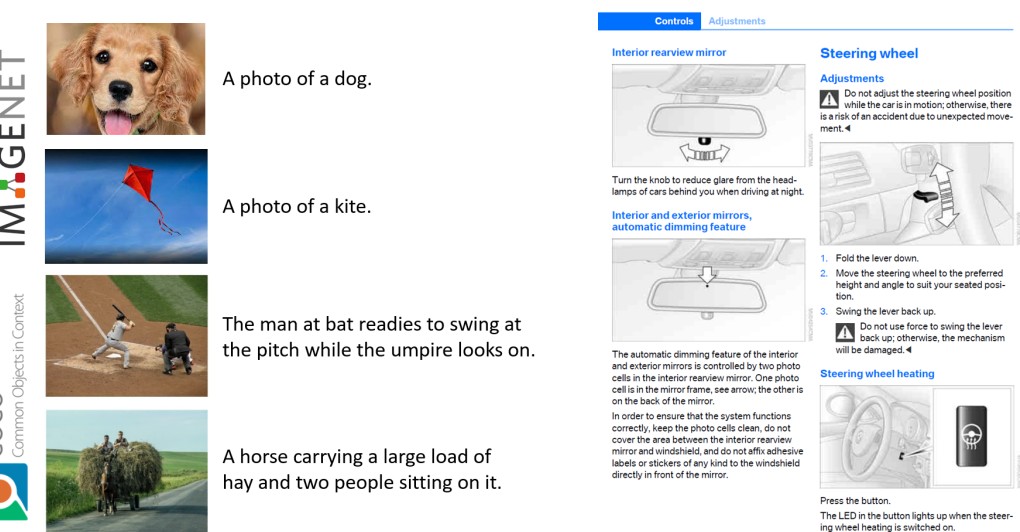

Figure 1: We introduce FETA, a novel dataset and benchmark for evaluating and improving Foundation (V&L) Models performance on expert data tasks. In contrast to mainstream benchmarks used to evaluate FMs, FETA does not focus on common objects captured with consumer cameras (left), instead providing a completely automatic pipeline for extracting (mostly other visual domains) expert data from publicly available technical and other documentation. Also, as opposed to original CLIP, FETA's MIL-CLIP method can learn from multiple-hypothesis data automatically extracted from complex document's pages (see right) comprised of multiple images and texts without apparent 1:1 association.

data may not be present in the web-crawled internet-scale datasets [16, 45, 50] used to train FMs; (ii) even if specialized data is present, it is deep in the long-tail of the data distribution statistics, meaning that due to the limited capacity, or the information bottleneck [57], of the FM models, useful representation features for this data are not significant in the FMs' learned representation space; (iii) commonly, a large domain gap exists between natural image common-objects biased data and the accompanying text used for FM training and sketch-like / synthetic / non-consumer-camera imagery commonly appearing in expert data scenarios. This suggests that to be utilized for expert data applications, FMs need to be tuned to better represent this data, driving the under-represented features that are necessary for such data to emerge. But how can one analyze and tune for such effects?

Our proposed Foundation models for Expert Task Applications (FETA) benchmark and dataset is intended to bridging the gap between the 'common object' oriented benchmarks (e.g. ImageNet) currently used to evaluate V&L FM performance and more complex specialized objects (e.g. an engine diagram, or a car mechanical part) typical to many real-world applications targeting expert tasks. To the best of our knowledge, the FETA is the first benchmark aiming at evaluating and improving the FMs performance in the expert data domains. The first version of FETA focuses on Text-to-Image (T2I) & Image-to-Text (I2T) retrieval in technical documentation, specifically diverse car service manuals from multiple manufacturers, and sales (currently IKEA annual) catalogues. Our proposed automatic annotation process is general and can support ingestion of any variety of programmatic PDF documents with illustrations, making FETA easily extensible to additional expert tasks and content domains, either by us (in future versions of FETA) or by other members of the community. The FETA is also equipped with our proposed method for automatic extraction of text-image pairs both for fine-tuning the models as well as for automatic performance evaluation. Our method is based on establishing multiple-hypothesis text-image correspondence via co-location of images and surrounding text on the pages of the processed PDFs. Although completely non-curated, we show how comparative metrics established by our proposed automatic annotation technique translate consistently to a metric established via manually curated ground truth data, thus indicating the utility

of the proposed automatic metric which, as explained above, effortlessly extends to any arbitrary expert tasks and content domains we expect to be added to FETA in the future. Finally, we provide a large set of interesting baselines on our collected FETA data including popular off-the-shelf FMs, various methods for finetuning FMs on the train set of FETA, as well as some interesting application of a combination of Locked and Multiple-Instance-Learning fine-tuning schemes demonstrating significantly superior performance on both automatic as well as manually curated metrics of FETA, paving the way to real practical applications of the proposed fine-tuning techniques.

**To summarize**, our key contributions are as follows: (i) We propose *Foundation models for Expert Task Applications* (FETA) - first of its kind dataset and benchmark - targeting the evaluation and improvement of the Foundation Models performance on expert data domain tasks prevalent in real-world applications; (ii) We propose and release an automatic text-image pairs extraction pipeline fitting any collection of illustrated programmatic PDFs or even broader documents data, making our proposed FETA easily extensible to new content domains and expert data applications drawing from this abundant source of *expert* V&L data; (iii) We propose an automatic evaluation metric for FMs on expert tasks using our proposed data extraction pipeline and show this metric leads to consistent models relative performance comparisons to the ones resulting from a manually curated metric (also released as part of FETA); (iv) We provide a large collection of baselines on FETA including out-of-the-box FMs, FETA tuned-FMs, and a novel combination of Low Ranked Adaptor finetuning using Multiple Instance Learning schemes reaching the best performance by a large margin; (v) Our findings corroborate our proposition that out-of-the-box FMs performance drops significantly when moving away from common object benchmarks and entering expert domains, underlining the value of our proposed FETA benchmark for future research towards paving the way to real-world practical applications of FMs in expert domains. The FETA dataset is available for download here[1]. The code is abailable at `https://github.com/alfassy/FETA`

## 2 Dataset Collection

### 2.1 Source and Description

Documents are a natural data-source to find text-image pairs, since images in documents have either captions or at the minimum are related to their surrounding textual content. In order to obtain real-world images, and not schematics or scientific figures, we chose documents related to consumer goods such as product catalogues and manuals. Such documents typically provide both images of the product and a textual description. This specific data was chosen due to several important criteria. First, the data is abundant with a large variety of text and images. Second, the data includes images which are not "natural" domain and belong to the long-tail distribution of the training data for the FMs. Finally, when collecting the data we considered the legal and privacy issues such that the data is freely available without any legal claims limiting its distribution. We downloaded 349 car service manuals from `https://www.workshopservicemanual.com/` each comprising 20 to 1602 pages. The documents were then processed, such that all text and images were automatically extracted. In the following sections we will describe in detail the automatic processing and annotation flow. Additionally, FETA also includes IKEA yearly catalogues data. The data was published by [66] for semantic based sentence recognition in images, the data is available for download[2]. The IKEA data was processed identically to the car-manuals dataset and is detailed in the supplementary material.

### 2.2 Data Conversion

The product-related source documents are in PDF format, which is notoriously hard to extract data from. In the past years, tools[3] and cloud-services[4] have been developed to convert PDF documents [3, 54] to JSON semantically, meaning that structural elements of the document (e.g. title, paragraphs, section-headers, tables, images, etc) are extracted semantically and easily accessible in the final JSON document. This semantic conversion is achieved by leveraging pre-trained ML methods for Layout-Segmentation [33, 42] & Table-Understanding [39]. In the Layout-Segmentation, document components such as text-blocks, tables and images are visually identified using state-of-the-art object-detection algorithms, which provide bounding-boxes for structural elements on each

---

[1]`https://ai-vision-public-datasets.s3.eu.cloud-object-storage.appdomain.cloud/FETA/feta.tar.gz`

[2]`https://github.com/ivc-yz/SSR`

[3]`https://github.com/DS4SD/deepsearch-toolkit`

[4]`https://deepsearch-experience.res.ibm.com`

Table 1: **Dataset Statistics**. In total, the first version of FETA contains around 56K extracted images and around 89K pieces of extracted image related text. Additional statistics are available in Section 2 of the supplementary material.

| Manufacturer | Docs | Avg. Pages/Doc | Avg. Images/Doc | Avg. Texts/Doc |
|---|---|---|---|---|
| Nissan | 275 | 149 | 138 | 249 |
| Toyota | 24 | 107 | 122 | 180 |
| Mazda | 9 | 149 | 413 | 657 |
| Chevrolet | 31 | 169 | 92 | 128 |
| Renault | 10 | 169 | 385 | 596 |
| Entire data Avg. | 349 | 149 | 147 | 254 |

page. The latter allows us to geometrically link images to text via a heuristic geometric closeness relationship. The output JSON undergoes extended post processing, aiming to reduce parsing noise. For example: merge spatially close texts by finding connected components in a graph made by the texts. More information available in Section 1 of the supplementary material.

### 2.3 Automatic Annotation

In the spirit of Multiple Instance Learning (MIL), we defined the automatic matching of each extracted image with a set of up to five pieces of texts from the **same page**. Every image was paired with the most probable text block from the left, right, top, and bottom of the figure, when available. We also selected a text box if it was overlapping with the image. We found that in the majority of the cases, at least one of these blocks of text is related to the image. This inherently creates the many-to-many MIL scenario where each image is associated with multiple text instances and vice-versa. We further found that in some cases, an image can appear in several places within the document. Since our automatic annotation is based on the co-location of the image and text within the same page, these cases can hurt retrieval metrics when disregarded. We thus applied a filtering process to merge all occurrences of the same image within a single document (see Section 1.2 of the supplementary material).

### 2.4 Manual Annotation

Since both training and test data were automatically annotated, we chose to manually annotate a small subset of the data in order to validate the results. For this manually annotated set we randomly selected 15 documents and manually paired a single image with a single text within every page of each document. The manual annotation was done using the annotation tool of the DeepSearch cloud service, based on the automatic extraction of images and texts. As can be seen from Table 2 and Table 3, the results on the automatic and manual annotated set are highly correlated, thus strengthening the validity of our proposed automatic annotation for testing, and underlining the scalability of our approach in terms of adding data and future inclusion of additional expert domains.

### 2.5 Statistics

The **Car-Manuals** dataset consists of a total of 349 PDF documents from 5 car manufacturers, namely Nissan, Toyota, Mazda, Renault, Chevrolet. Table 1 details the statistics of the dataset by manufacturer. The **IKEA-catalogues** dataset contains 26 documents with 7366 pages total, approximately 9574 images and 23927 texts automatically extracted from those pages. More details on the IKEA-catalogues dataset, as well as analysis of the performance of our rich set of baselines on that dataset and further data statistics is provided in Section 2 of the supplementary material.

## 3 Baselines

### 3.1 Background

Our main out-of-the-box FM baseline is the most widely used and readily available V&L FM, CLIP [45]. CLIP was trained using a contrastive loss applied to the similarity of the textual and visual features of all image-text pairs within each batch. This simple yet effective method has proven to work fantastically on natural images when supplied with a huge amount (400M) of image-text pairs collected from the Web. However, as we show in our experiments (Table 2), CLIP's performance on the document data based expert task is far from sufficient. This underlines the need to fine-tune models such as CLIP on expert tasks in order to adapt the model to practical use in expert applications. But what is the best and most scalable way to fine-tune in this case? In the case of the automatic

document data annotation, where there are no image-text pairs but rather sets of text associated to each image and sets of images associated with each text, we argue that the original contrastive loss can not be used, and the proposed MIL variants, inspired by [37] from the video domain, should be used instead. Additionally, expert data is quite diverse and significantly small compared to the tremendous volumes of pre-training data used to make CLIP. We therefore also explore different constrained fine-tuning strategies based on encoder-locking ideas from [64].

**CLIP**. The CLIP model is comprised of an image encoder and a text encoder. Let $\mathcal{M}_I$ and $\mathcal{M}_T$ be the image and text encoders respectively. For a given image $I_i$, and a piece of text $T_i$, we define the image and the text embedding vectors as the outputs of $\mathcal{M}_I$ and $\mathcal{M}_T$, respectively:

$$x_i = \mathcal{M}_I(I_i), \qquad y_i = \mathcal{M}_T(T_i) \tag{1}$$

Standard supervised learning assumes that the samples and targets are paired, $\{x_i, y_i\}_{i=1}^N$, where N is the size of the dataset. For a given batch of samples, $B$, the standard CLIP loss is a cross-entropy loss defined as:

$$\mathcal{L}_{CLIP} = -\frac{1}{2B}\left(\sum_i^B \log \frac{\exp(x_i^T y_i/\sigma)}{\sum_{j=1}^B \exp(x_i^T y_j/\sigma)} + \sum_i^B \log \frac{\exp(y_i^T x_i/\sigma)}{\sum_{j=1}^B \exp(y_i^T x_j/\sigma)}\right) \tag{2}$$

where $\sigma$ is a normalization factor, often set as a learned parameter.

We propose an extension to the CLIP contrastive loss, adapting it to a MIL setting where we know that at least on of the texts is a positive match to the image and vice versa.

### 3.2 MIL-CLIP

The MIL setting, relaxes the paired assumption and defines a "bag" of $M$ targets $\{y_i^m\}_{m=0}^M$ such that at least one of the targets (e.g. texts) is a positive match to the sample (e.g. image). This weak annotation aligns perfectly with our automatic annotation framework.

There are several ways to modify the original loss (Eq. 2) to the MIL setting. Next, we will present a few plausible baselines.

**MIL-Max**. A simple yet effective method for MIL is by selecting the positive example as the maximum value over the bag of labels. Defining $\hat{m}_i = \arg\max_m x_i^T y_i^m$:

$$\mathcal{L}_{MAX} = -\frac{1}{B}\sum_i^B \log \frac{\exp(x_i^T y_i^{\hat{m}_i}/\sigma)}{\exp(x_i^T y_i^{\hat{m}_i}/\sigma) + \sum_{j\neq i}^B \sum_m \exp(x_i^T y_j^m/\sigma)}$$
$$-\frac{1}{B}\sum_i^B \max_q \log \frac{\exp(y_i^{qT} x_i/\sigma)}{\exp(y_i^{qT} x_i/\sigma) + \sum_{j\neq i}^B \sum_m \exp(y_i^{mT} x_{/}\sigma)} \tag{3}$$

**MIL-SoftMax**. A small modification to the MIL-Max loss is replacing the maximum with a SoftMax weighted average of the nominator of the loss function. For convenience we first define the SoftMax weights with scaling factor $\sigma_{sm}$ as:

$$S_i^m = \frac{\exp(x_i^T y_i^m/\sigma_{sm})}{\sum_{n=1}^M \exp(x_i^T y_i^m/\sigma_{sm})} \tag{4}$$

and define the MIL-SoftMax variant as:

$$\mathcal{L}_{SM} = -\frac{1}{B}\sum_i^B \log \frac{\sum_m S_i^m \exp(x_i^T y_i^m/\sigma)}{\sum_m S_i^m \exp(x_i^T y_i^m/\sigma) + \sum_{j\neq i}^B \sum_m \exp(x_i^T y_j^m/\sigma)}$$
$$-\frac{1}{B}\sum_i^B \max_q \log \frac{\exp(y_i^{qT} x_i/\sigma)}{\exp(y_i^{qT} x_i/\sigma) + \sum_{j\neq i}^B \sum_m \exp(y_i^{mT} x_{/}\sigma)} \tag{5}$$

**MIL-NCE**. Recently the MIL-NCE [37] approach was proposed for visual representation learning from uncurated videos. We adapt the MIL-NCE loss to fit the clip, contrastive loss as follows :

$$\mathcal{L}_{NCE} = -\frac{1}{B}\sum_i^B \log \frac{\sum_m \exp(x_i^T y_i^m/\sigma)}{\sum_{j=1}^B \sum_m \exp(x_i^T y_j^m/\sigma)} \tag{6}$$

Following the ablation study presented in the supplementary material, unless stated otherwise we chose the MIL-NCE version in all our experiments.

### 3.3 CLIP-LoRA

LoRA [17] proposed Low-Rank Adapters for large language models. LoRA locks the original weights of a pretrained model and adds trainable low rank residual adapters to different model layers. LoRA build upon the observation that in many cases learned layer weight matrices are in fact of low-rank, hence adapting them with a low-rank constraint on the change leads to good results also increasing efficiency and reducing over-fitting. We defer in our use of LoRA adapters both from the original work of [17], as well as from the only reported use of this tool for V&L models so far: [55]. As opposed to [17] and [55] that injected LoRA only into query/value projections of transformer MHSA blocks ([17]) or introduced them only to the text encoder ([55]), we found that also using LoRA weights in all nn.Conv2d, nn.Linear and nn.Embedding layers, results in significantly bigger performance gains. Additionally, we also apply LoRA outside just the transformer models - also to the CLIP ResNet50 backbone image encoder where applicable.

### 3.4 Implementation Details

We base our code on the open project [18] `https://github.com/mlfoundations/open_clip`. All our experiments used ResNet50 backbone models using the original CLIP [45] pre-trained models (400M image-text pairs). All fine-tuning experiments were run with 5e-05 learning rate, 64 batch size, and 20 epochs, using PyTorch DDP. We provide our code, including our automatic data annotation and all the baselines, in the supplementary and will release it upon acceptance. All the models were trained using either an Nvidia A100 or V100 GPU, with 8 GPUs per experiment.

## 4 Experiments

### 4.1 Data Splits

In all of the following experiments the data was split into five folds for each manufacturer and the results presented are an average over the results of the five-fold training and testing regime. The results are further averaged across manufacturers. Our folds are splitting on complete documents, not on document pages, so pages from the same document never appear in both train and test. We next present a detailed set of baseline experiments under four different settings: (i) *Many-Shot*: training on four folds of a *Nissan manufacturer*, as it contains the largest collection of documents, and testing on the remaining fifth fold; (ii) *Zero-Shot*: training on all data of all but one manufacturer, testing on all the data of the left-out manufacturer; . (iii) *One-shot*: similar to Zero-Shot but adding one document of the left-out manufacturer, testing on the remaining data of the left-out manufacturer, this is repeated five times with a different document each time; (iv) *Few-shot*: similar to One-Shot but adding one **fold** of the left-out manufacturer, testing on the remaining folds of the left-out manufacturer, this is repeated five times with a different document each time.

### 4.2 Baseline Methods

In addition to our MIL-CLIP method we evaluate three simple CLIP-based baselines: (i) *CLIP*: We test the pre-trained CLIP400M model without any further training. (ii) *Concatenate*: During training we concatenate all texts from the MIL "bag" into one long text and set it as the positive example and train using the original CLIP loss (Eq. 2). (iii) *Choose-One*: During training we randomly select one of the texts from the MIL "bag" as the positive example and train using the original CLIP loss (Eq. 2). For both Concatenate and Choose-One we test both Locked and non-Locked variants. Additional information on the evaluation of FLAVA [52], ALBEF [31], and VilT [25] is available in Section 4.3 of the supplementary material.

### 4.3 Results

Table 2 presents the comparison of three baseline methods (also with or without Locking) under four different data split settings. These empirical results lead to several interesting conclusions. **First**, the CLIP model under-performs with respect to all the fine-tuning methods in the Zero-Shot and other settings, strengthening our hypothesis that FMs indeed need to be fine-tuned for expert domain (practical) applications such as explored in FETA, and their massive-scale pre-training is not sufficient for this tasks on its own. **Second**, fine-tuning using automatically collected V&L annotations induces significant performance improvements in many cases, especially in the Many-Shot case, which is arguably the most practical scenario, as the annotations are automatic hence the train data can scale easily with adding more documents. This further highlights the benefit of automatic annotation pipeline proposed in FETA for supporting low annotation cost adaptation of V&L models to expert

Table 2: **Main Results:** Image-to-Text and Text-to-Image retrieval accuracy for different baselines under three different data-split settings. Our baselines include out-of-the-box CLIP (additional FM results provided in Supplementary), several variants of its non-MIL and MIL fine-tuning variants. Our experimental settings include Many-Shot (train and test on same manufacturer data with lots of train samples), Zero-Shot (train and test on different manufacturers data), One-Shot (like Zero-Shot, but include a single document of the tested manufacturer in training), and Few-Shot (like Zero-Shot, but include a single fold of the tested manufacturer data in training). Results are averaged across manufacturers. All the experiments were performed on the automatically annotated data using five-folds and (naturally) without any overlap between train and test (on the level that pages of the same document *never appear* in both train and test). The "Locked" column refers to versions trained with locked (frozen) parameters of the image encoder $\mathcal{M}_I$. For reference, in FETA - the random chance probabilities for guessing the correct text match or the correct image match are 1.14% and 0.67% respectively. Numbers in **bold**/blue mark the best and second-best results, respectively.

| | Name | Locked | Image-to-Text | | | Text-to-Image | | |
| --- | --- | --- | --- | --- | --- | --- | --- | --- |
| | | | Rec@1 | Rec@5 | Rec@10 | Rec@1 | Rec@5 | Rec@10 |
| Zero-Shot | CLIP [45] | | 9.7% | 26.6% | 38.1% | 10.1% | 26.7% | 39.5% |
| | FLAVA [52] | | 4.0% | 16.6% | 29.7% | 5.5% | 19.8% | 34.5% |
| | VilT [25] | | 2.9% | 11.5% | 22.0% | 3.5% | 14.3% | 26.7% |
| | ALBEF [31] | | 3.9% | 16.6% | 26.7% | 4.4% | 18.4% | 31.2% |
| | Concatenate | | 6.5% | 20.4% | 31.5% | 7.1% | 25.0% | 38.4% |
| | Concatenate | ✓ | 9.4% | 25.0% | 36.7% | 8.1% | 24.0% | 37.6% |
| | Choose-One | | 10.7% | 27.6% | 39.9% | 9.3% | 28.1% | 41.8% |
| | Choose-One | ✓ | 10.40% | 26.7% | 39.3% | 9.20% | 25.6% | 37.9% |
| | CLIP-MIL | | 10.5% | **34.0%** | **48.5%** | **11.7%** | **32.9%** | **47.9%** |
| | CLIP-MIL | ✓ | **11.0%** | 29.2% | 40.0% | 9.7% | 28.1% | 40.6% |
| One-Shot | CLIP [45] | | 9.7% | 26.6% | 38.1% | 10.1% | 26.7% | 39.4% |
| | FLAVA [52] | | 4.0% | 16.6% | 29.7% | 5.5% | 19.8% | 34.5% |
| | VilT [25] | | 2.9% | 11.5% | 22.0% | 3.5% | 14.3% | 26.7% |
| | ALBEF [31] | | 3.9% | 16.6% | 26.7% | 4.4% | 18.4% | 31.2% |
| | Concatenate | | 10.3% | 27.3% | 39.2% | 9.5% | 27.0% | 40.8% |
| | Concatenate | ✓ | 8.7% | 24.4% | 37.0% | 7.9% | 25.1% | 38.5% |
| | Choose-One | | 10.3% | 27.2% | 39.5% | 9.4% | 27.5% | 41.6% |
| | Choose-One | ✓ | 10.4% | 28.0% | 39.9% | 8.9% | 25.5% | 37.9% |
| | CLIP-MIL | | 11.0% | **30.3%** | **43.2%** | 9.9% | 27.9% | 40.9% |
| | CLIP-MIL | ✓ | **11.9%** | **30.3%** | 42.5% | **10.9%** | **29.4%** | **43.2%** |
| Few-Shot | CLIP [45] | | 8.6% | 25.6% | 37.2% | 9.2% | 24.3% | 36.6% |
| | FLAVA [52] | | 3.9% | 16.7% | 30.1% | 5.2% | 18.5% | 32.7% |
| | VilT [25] | | 2.8% | 11.1% | 21.6% | 3.2% | 13.3% | 25.4% |
| | ALBEF [31] | | 3.9% | 13.1% | 25.5% | 4.2% | 17.4% | 29.6% |
| | Concatenate | | 9.0% | 26.5% | 40.3% | 10.3% | 29.6% | 44.7% |
| | Concatenate | ✓ | 8.4% | 24.5% | 38.5% | 10.5% | 29.0% | 41.8% |
| | Choose-One | | 11.2% | 30.5% | 44.2% | 13.1% | 31.3% | 46.4% |
| | Choose-One | ✓ | 11.6% | 31.5% | 44.7% | 11.6% | 29.0% | 44.0% |
| | CLIP-MIL | | **14.1%** | **36.7%** | **48.9%** | **15.0%** | **35.2%** | **50.0%** |
| | CLIP-MIL | ✓ | 13.8% | 33.7% | 47.5% | 11.6% | 33.0% | 47.0% |
| Many-Shot | CLIP [45] | | 13.8% | 31.2% | 41.6% | 13.6% | 36.4% | 50.7% |
| | FLAVA [52] | | 4.2% | 16.1% | 27.8% | 6.6% | 24.4% | 41.0% |
| | VilT [25] | | 3.1% | 13.3% | 23.4% | 4.4% | 18.3% | 32.0% |
| | ALBEF [31] | | 3.8% | 15.8% | 26.6% | 5.5% | 22.4% | 37.5% |
| | Concatenate | | 18.4% | 37.8% | 49.9% | 15.9% | 42.1% | 58.3% |
| | Concatenate | ✓ | 20.7% | 39.7% | 51.3% | 16.2% | 41.2% | 56.2% |
| | Choose-One | | 24.5% | 49.9% | 62.2% | 21.2% | 52.3% | 67.2% |
| | Choose-One | ✓ | 27.7% | 52.7% | 64.1% | 21.6% | 52.1% | 66.5% |
| | CLIP-MIL | | 32.6% | 56.2% | **66.7%** | **27.8%** | **59.0%** | **72.3%** |
| | CLIP-MIL | ✓ | **34.5%** | **56.8%** | 66.1% | 27.2% | 57.9% | 70.7% |

Table 3: **Results on manually curated data:** Image-Text retrieval accuracy results. Models trained on automatically annotated data in the same way as in table 2 excluding the manually annotated docs. Models are tested on a small manually annotated subset of the data in order to verify the results of Table 2 that were measured using our automatic annotation. Numbers in **bold**/blue mark the best and second-best results, respectively.

| | Name | Locked | Image-to-Text Rec@1 | Rec@5 | Rec@10 | Text-to-Image Rec@1 | Rec@5 | Rec@10 |
|---|---|---|---|---|---|---|---|---|
| Zero-Shot | CLIP [45] | | 14.3% | 39.3% | 55.6% | 14.7% | 36.4% | 57.0% |
| | Concatenate | | 14.2% | 40.2% | 61.6% | 11.5% | 33.6% | 51.9% |
| | Concatenate | ✓ | 9.3% | 35.4% | 58.5% | 11.7% | 31.5% | 51.5% |
| | Choose-One | | 9.8% | 39.1% | 59.9% | 13.3% | 38.2% | 59.9% |
| | Choose-One | ✓ | 12.8% | 40.5% | 59.0% | 14.2% | 41.9% | 60.0% |
| | CLIP-MIL | | **14.4%** | 40.1% | **67.8%** | **16.2%** | 40.4% | **62.6%** |
| | CLIP-MIL | ✓ | 13.9% | **41.0%** | 65.0% | 15.1% | **43.2%** | 60.8% |
| One-Shot | CLIP [45] | | 14.3% | 39.3% | 55.6% | 14.7% | 36.4% | 57.0% |
| | Concatenate | | 15.7% | 41.9% | 63.3% | 12.1% | 39.1% | 60.8% |
| | Concatenate | ✓ | 12.4% | 37.7% | 54.1% | 13.0% | 35.3% | 54.9% |
| | Choose-One | | 12.7% | 40.8% | 60.4% | 12.4% | 37.8% | 63.2% |
| | Choose-One | ✓ | 12.4% | 38.8% | 62.5% | 14.2% | 39.5% | 62.1% |
| | CLIP-MIL | | **16.1%** | **42.7%** | **66.2%** | 14.6% | **43.6%** | **64.0%** |
| | CLIP-MIL | ✓ | 15.8% | 39.9% | 62.6% | **14.9%** | 41.6% | 62.5% |
| Few-Shot | CLIP [45] | | 12.8% | 37.3% | 51.7% | 12.5% | 32.1% | 53.0% |
| | Concatenate | | 12.0% | 38.3% | 58.9% | 11.0% | 32.5% | 54.7% |
| | Concatenate | ✓ | 8.2% | 33.0% | 55.4% | 8.8% | 28.9% | 51.3% |
| | Choose-One | | 11.0% | 39.8% | 60.9% | 12.5% | 36.4% | 60.1% |
| | Choose-One | ✓ | 9.8% | 37.0% | 59.5% | 10.1% | 35.7% | 59.7% |
| | CLIP-MIL | | **13.4%** | **41.3%** | **61.9%** | 12.7% | 38.2% | 60.5% |
| | CLIP-MIL | ✓ | 12.9% | 38.4% | 60.3% | **13.2%** | **38.6%** | **61.6%** |
| Many-Shot | CLIP [45] | | 20.0% | 47.2% | 71.3% | 23.7% | 53.4% | 72.9% |
| | Concatenate | | 36.8% | 66.6% | 84.1% | 24.5% | 56.5% | 80.5% |
| | Concatenate | ✓ | 29.9% | 61.6% | 81.1% | 28.8% | 56.4% | 76.3% |
| | Choose-One | | 39.0% | 70.1% | 83.1% | 34.0% | 65.0% | 84.7% |
| | Choose-One | ✓ | 34.6% | 70.6% | 83.5% | 33.7% | 68.3% | 82.6% |
| | CLIP-MIL | | **43.1%** | 71.4% | 83.8% | **40.7%** | 67.6% | **86.5%** |
| | CLIP-MIL | ✓ | 43.0% | **74.8%** | **85.7%** | 43.5% | **70.2%** | 85.3% |

domains defined by corpora of documents with illustrations. **Third**, training with the MIL paradigm consistently boost performance with respect to other (non-MIL) baselines indicating the utility of using MIL and variants. **Fourth**, locked image encoder variants demonstrate interesting trade-offs with unlocked ones in different scenarios. We have further evaluated this in a more thorough ablation study of this aspect in Section 4.4, also discovering the benefit of very interesting intermediate locking options using low-rank residual adapters tuning, constituting a very exciting direction for future work. **Fifth**, overall performance levels of all baselines still leave a lot of room for improvement for future research towards practical application of FMs to (abundant) real-world expert domain application tasks. Furthermore, Table 3 validates the results in Table 2 by measuring the performance of the same baselines using a manually curated annotated documents set. The results are validated by observing the consistent performance trends between the baselines in the two tables.

## 4.4 Additional Ablation Study - Parameter Locking

Following [64], we evaluated the performance on our CLIP-MIL method under several parameter-locking as well as "intermediate" states. We refer to locked parameters as parameters that do not change during training. The five different options are: (i) *Unlocked:* Let both $\mathcal{M}_I$ and $\mathcal{M}_T$ train during fine-tuning. (ii) *Locked Image:* Lock $\mathcal{M}_I$ and only let $\mathcal{M}_T$ train. (iii) *Locked Text:* Lock $\mathcal{M}_T$ and only let $\mathcal{M}_I$ train. (iv) *Locked*:* Lock both $\mathcal{M}_I$ and $\mathcal{M}_T$ except the last "text projection" layer in $\mathcal{M}_T$. (v) CLIP-LoRA as detailed in CLIP-LoRA sub section. Table 4 clearly shows the trade-offs

Table 4: **Parameter Locking Ablation:**. This table explores different variants of locking the model parameters during MIL finetuning in the Many-Shot setting. We test locking the image encoder, the text encoder, or both excluding the text projection layers (Locked*) and the use of Low-Rank Adapters (LoRA) [17]. We show the interpolation between the Unlocked (rank r = 512) and the Locked Image (rank r = 0) variants by changing the rank of the added residual adapters weight matrices. This exploration clearly shows the trade-offs between locking and unlocking the image encoder MI , with up to 2.1% relative improvements in some cases. Numbers in **bold**/blue mark the best and second-best results, respectively.

| Baseline | Locking | Image-to-Text | | | Text-to-Image | | |
|---|---|---|---|---|---|---|---|
| | | Rec@1 | Rec@5 | Rec@10 | Rec@1 | Rec@5 | Rec@10 |
| CLIP-MIL | Unlocked | 32.6% | 56.2% | 66.7% | 27.8% | 59.0% | 72.3% |
| | Locked Image | 34.5% | 56.8% | 66.1% | 27.2% | 57.9% | 70.2% |
| | Locked Text | 30.6% | 54.4% | 64.6% | 27.8% | 59.0% | 71.6% |
| | Locked* | 30.1% | 52.4% | 63.1% | 25.0% | 55.7% | 69.1% |
| | LoRA[17] r=4 | 33.8% | 57.3% | 67.6% | 28.9% | 61.4% | 74.1% |
| | LoRA[17] r=32 | **35.6%** | **58.3%** | **68.1%** | 30.7% | **62.6%** | **74.6%** |
| | LoRA[17] r=256 | 35.5% | 57.7% | 67.8% | **30.8%** | 62.4% | 74.4% |

between locking and unlocking the image encoder $\mathcal{M}_I$, with up to $2.1\%$ relative improvements in some cases. More importantly this could be further significantly improved by low-rank intermediate variants with up to 2-3% additional improvement. Notice that no parameters are added as these adapters are only used for training and are fully collapsed into the model parameters at inference time. We believe that this shows that some adaptation to the

## 4.5 IKEA results

Table 5 presents the results for the proposed baselines trained and tested on the IKEA dataset. For a full review of IKEA dataset see section 2 in supplementary. For a discussion about the results, see section 4.5 in the supllementary.

Table 5: **Results on IKEA dataset** using 5-fold cross-validation protocol on the entire IKEA US early manuals data. MIL based baselines obtain significant advantages over other baselines. Numbers in **bold** mark the best results while numbers in blue mark the second-best.

| | Name | Locked | Image-to-Text | | | Text-to-Image | | |
|---|---|---|---|---|---|---|---|---|
| | | | Rec@1 | Rec@5 | Rec@10 | Rec@1 | Rec@5 | Rec@10 |
| All-Data | CLIP [45] | | 22.9% | 43.3% | 54.2% | 25.5% | 46.8% | 59.5% |
| | Concatenate | | 6.7% | 13.7% | 18.2% | 13.2% | 27.0% | 35.9% |
| | Concatenate | ✓ | 8.1% | 15.6% | 20.6% | 14.0% | 26.9% | 35.3% |
| | Choose-One | | 15.1% | 30.2% | 38.5% | 17.9% | 36.2% | 46.4% |
| | Choose-One | ✓ | 14.1% | 28.0% | 35.3% | 16.4% | 32.3% | 41.8% |
| | CLIP-MIL | | **26.8%** | **47.7%** | **57.8%** | **30.1%** | **54.4%** | **66.2%** |
| | CLIP-MIL | ✓ | 24.4% | 44.4% | 54.7% | 27.0% | 49.9% | 60.5% |

## 5 Related Work

**Vision and Language**. Many studies have recently addressed the problem of vision and language on a broad scale. Some of them focused more on text-image, such as [1, 14, 21, 30, 31, 32, 45, 52, 60, 62, 63], while others explored text-conditional image generation [40, 47, 49]. Other approaches learn strong representations from video-textual descriptions [37, 38] with or without the need for any manual annotation. The goal of these works is to learn foundational language and vision representations that are required for language and vision understanding. Unlike these works, we demonstrate here that even strong models are incapable of performing basic retrieval capabilities in technical documentation as humans do, such as diverse car service manuals and sales catalogs.

**Multiple Instance Learning**. Over the years, multiple instance learning methods have been applied to a variety of weakly supervised problems including: images [20, 41, 44, 53, 61, 67], videos [5, 10, 29,

36, 37, 51]. Typically, MIL methods are using different principles such as max-pooling [13], support vector machine [2], discriminative clustering [4], or even attention-based neural networks [19]. In this work, we present MIL-CLIP, an approach that combines the standard contrastive learning from CLIP [45] with multiple instance learning [13, 23, 34]. We demonstrate how this combination leads to the best performance and allows for practical applications of FMs in expert domains

**Image-Text Retrieval**. Image-text retrieval has been a long and well-known task with real-life applications. The two main and dominate tasks are: *image retrieval* and *text retrieval*, depending on which modality is used as the retrieved target. Previous works embedded the image and text features into a joint embedding space to calculate the similarities between them. Most of these works were trained by ranking loss [26, 58, 59], while more recent architectures and pre-training approaches [11, 45, 65] have demonstrated the potential of transformer-based models and contrastive objectives to learn image representations from text. In this work, we release a dataset containing manuals of cars and sale catalogs, showing that even large models cannot perform well on retrieval tasks, such as *image retrieval* and *text retrieval*. We hope it would pave the way for real practical applications of FMs in expert domains.

**Technical and expert domains with non-natural image data**. While the majority of CV literature focuses on natural images and common objects, some works have extended CV and V&L techniques to technical and expert domains (e.g., localization for autonomous driving [7], etc.). These works can be divided into works with mostly (i.) uni-modal focus, with such tasks as deep normal prediction in design sketches [15], image-to-image retrieval in patents [27, 43] (interestingly [43] also show that textual side-information can facilitate retrieval), scientific-figures classification [22], or text-to-text generation for patent claims [28]; (ii.) multi-modal works focusing on image+text reasoning tasks such as VQA on figures and InfoGraphics [9, 35, 24]. In contrast, FETA focuses on a more direct multi-modal V&L evaluation of out-of-the-box and fine-tuned, large-scale pre-trained, V&L models using text-to-image and image-to-text retrieval tasks, which are better aligned with the commonly used contrastive objectives used to pre-train V&L models. Moreover, FETA is open-ended, offering a convenient ingestion pipeline for producing automatic annotation and for the evaluation and fine-tuning of expert domains available as document corpora. This pipeline enables relatively straightforward future extensions to expert domains such as patents, figures and info-graphics.

## 6   Conclusion

We have proposed the first of its kind *Foundation models for Expert Task Applications* (FETA) benchmark and dataset focused on evaluating Foundation Models on expert data tasks. In our first release, FETA focuses on expert data from technical and other documents. It is accompanied with an automatic data extraction pipeline allowing for easy extension of the benchmark to larger data scales, other expert domains, and additional visual modalities by ingesting public PDF documents – an abundant data resource. FETA is accompanied with an extensive set of baselines and ablations on different training setups and finetuning strategies allowing us to conclude that: (i) Although strong on benchmarks containing common objects captured with consumer cameras, FMs still struggle with expert domain data, both due to its natural domain gap as well as absence or statistical insignificance of such data in the distribution of the massive datasets used to pre-train FMs; (ii) While our baselines still leave a lot of room for improvement contingent on future research, as expected of any good and challenging benchmark, in some situations such as many-shot fine-tuning, our best baseline performance suggests a possibility of practical application; (iii) Our diverse experimental settings help establishing best practices for fine-tuning FMs under different data regimes, and our code is easily extendable to evaluate any arbitrary FM in a similar collection of settings; (iv) Our automatic annotation pipeline and associated automatic performance metric lead to similar conclusions with regards to relative performance comparisons between different models and fine-tuning strategies, as the metric computed on the manually curated data, once again suggesting the scalability of the proposed approach to grow to larger data and additional expert tasks.

**Limitations & Future Work.** While the first version of FETA includes close to 150K images and texts, it is still a drop in the ocean of available technical documentation and other documents available for yet unexplored set of different expert V&L data domains. Luckily, FETA's automatic data extraction and annotation pipeline allows to scale FETA easily. Future work includes expanding FETA to additional domains and continually evaluating new FMs as they are released to the community.

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
