# OpenReview forum: "FETA: Towards Specializing Foundational Models for Expert Task Applications"
_NeurIPS.cc/2022/Track/Datasets_and_Benchmarks — NeurIPS 2022 Datasets and Benchmarks _

### Official Review · Reviewer_XJma · 2022-07-21

**Rating:** 8
**Confidence:** 4
**Correctness:** I don't see any significant correctne…
**Clarity:** The submission is well-written and cl…

**Strengths:**

- The submission is clear and well-written.
- The benchmark highlights failure modes in vision and language foundational models.
- Many adaptation strategies are evaluated.
- The submission contains a high level of implementation details and provides reference code.

**Weaknesses:**

- Baseline results are very CLIP-centric, although results for FLAVA are presented in the supplementary material.
- Not a whole lot is said about the choice of evaluation metric in the main text. The submission would benefit from a paragraph justifying this choice.

**Additional Feedback:**

Overall this is a strong submission from my perspective. The two key aspects that would strengthen it would be to elaborate on the choice of evaluation metric and to expand the discussion of related work to draw parallels to other subfields in terms of generalizing to very different downstream tasks.

**Documentation:**

The submission provides sufficient details (URL, reference implementation, documentation) to support reproducibility.

**Ethics:**

I don't see any ethical concerns warranting further discussion.

**Relation To Prior Work:**

The submission's discussion of the foundational models, vision and language, multiple instance learning, and image-text retrieval literature is thorough.

The inability of foundational models to perform well on tasks way outside of the training distribution being one of the submission's core observations, it would be relevant to discuss how this echoes findings in other subfields such as cross-domain few-shot classification, domain adaptation and generalization, and transfer learning.

**Summary And Contributions:**

**UPDATE**: After reading all reviews and discussions, I remain positive about acceptance. I agree with other reviewers that the automated labeling procedure is a source of potential label noise in the dataset, but I believe the authors have taken reasonable steps to verify the annotations.

----------

The submission introduces a benchmark called "Foundational models for Expert Task Applications" (FETA) which evaluates the performance of foundational models (FMs) for vision and language on expert tasks whose data distributions are very different from the data distribution seen during training. More specifically, the benchmark focuses on matching illustrations to their textual description in technical documentation for car service manuals and IKEA yearly catalogues.

The PDF source documents are parsed using IBM's Deep Search tool to extract images and text, and images are automatically paired with up to five text snippets from the same page, resulting in images being associated with multiple pieces of text (and vice versa). A small subset of the data is also manually annotated to validate the correctness of the automatic annotation procedure.

The submission uses the pre-trained open-source CLIP model for baseline evaluation along with several extensions which adapt the model to the FETA data distribution in varying data regimes. To address the many-to-many nature of the benchmark's annotations, the paper proposes to consider multiple instance learning (MIL)-inspired loss formulations. Parameter-locking approaches are also investigated in an ablation study. Performance is measured in terms of recall@{1,5,10} for both image-to-text and text-to-image retrieval accuracy.

Results on the manually-annotated subset are shown to be consistent with results on the automatically-annotated data. CLIP without any adaptation is shown to underperform FETA's expert domain data, and conversely the adaptation strategies investigated improve model performance, especially in the case of MIL-inspired strategies.

---

> ### Author Response · Authors · 2022-08-18
> **Detailed Response to Reviewer XJma**
>
>
> We thank the reviewer for the constructive feedback. Following is our detailed response:
>
> -----
>
> > Baseline results are very CLIP-centric, although results for FLAVA are presented in the supplementary material.
>
> #####
>
> We thank the reviewer for the positive and constructive feedback. In the original submission we included CLIP-based and FLAVA baselines. These are the top-performing openly available V&L models at the time of the original submission, and to the best of our knowledge also until this time. We however adopt the reviewer's suggestion and are working on adding more openly available (older) V&L model baselines to FETA.
>
>
> -----
>
> > Not a whole lot is said about the choice of evaluation metric in the main text. The submission would benefit from a paragraph justifying this choice.
>
>
> ### Answer
>  We thank the reviewer for their important suggestion. We choose the text-to-image and image-to-text retrieval task for two main reasons:
>  1. This metric is directly aligned with the popular contrastive training objective used for most of V\&L models (e.g. CLIP or FLAVA) and as such should be their strongest suit. We however show that even under this metric CLIP underperforms on FETA expert tasks compared to its performance demonstrated for e.g. photos of common objects.
>
>  1. This metric is also possible to compute when regarding the automatic annotation process. In our automatic process, little is known a priori about the data. The assumption that co-location of text and images is strongly correlated with the semantics is a relatively general assumption. We show in our Result section 4.3 and in Table 3 that our choice of automatic metric is valid as it correctly predicts the trends for the same models and baselines performance as evaluated on the manually annotated and curated data. We further strengthen this claim in a new Section 2.1 of the revised supplementary material where we compare the overlap of the manual and the automatic annotation. We thank the reviewer for the insightful comment and will add a discussion on the metric selection to the manuscript.
>
> -----
>
> > The inability of foundational models to perform well on tasks way outside of the training distribution being one of the submission's core observations, it would be relevant to discuss how this echoes findings in other subfields such as cross-domain few-shot classification, domain adaptation and generalization, and transfer learning.
>
> ### Answer
>
> We thank the reviewer for this comment,  we will add such a discussion to the related works section by the end of the discussion period.

---

> ### Author Response · Authors · 2022-08-29
> **Additional V&L Methods and Discussion**
>
> We thank the reviewer for the constructive  and important comments and suggestions.
>
> As promised we added two new V&L methods in addition to CLIP and FLAVA to Table 2 of the revised manuscript namely VilT and ALBEF.
>
> Furthermore, as promised we have added a discussion on the choice of metric to Section 2.5 of the revised supplementary material
>
> We hope that these new changes are answer the reviewer’s concerns

---

### Official Review · Reviewer_rF7u · 2022-07-26
**A text-to-image and image-to-text retrieval benchmark to evaluate foundational models' performance on expert tasks such as car manuals.**

**Rating:** 6
**Confidence:** 4

**Strengths:**

1. This paper proposes a new retrieval dataset for the understanding of expert documents.
2. They propose an automatic text-image pairs extraction pipeline for any kind of PDF file.
3. With experiments in different settings, they show the effectiveness of Multiple Instance Learning on retrieval tasks.

**Weaknesses:**

1. This paper provides the statistics of the FETA including the number of documents for each car manufacturer and the average text/image numbers in these documents. Nevertheless, each car manufacturer may provide different content in their car manuals. The paper should give a more detailed analysis of the text and images in the documents.
2. Although the authors provide the IKEA-catalogues dataset and provide an experimental evaluation of it, they do not provide any analysis of it.
3. After downloading the dataset, the compressed file includes several directories of car manuals without any files in them. The compressed file also includes IKEA images but no texts. The supplementary material should provide a more clear guideline to use FETA.

**Additional Feedback:**

Nothing to add here. The sections above contain the main points I want to discuss.

**Clarity:**

The paper is fairly well written and contains several grammar and spelling errors. For example:

Line 97: comprising of -> comprising

Line 123: ... within the same page these cases...-> within the same page, these cases...

Line 153: stets -> sets

Line 232: evaluated the performance or ->  evaluated the performance on

Also, using questions in academic papers may not be common, and asking an open question is not appropriate.

eg. Line 45, Line 151

Further, when we mention the supplementary material in the main paper, it is better to provide the section number and the hyperlink.

**Correctness:**

The authors indicate that there is a large natural domain gap between their expert domain data and common object data. However, based on their ablation study of parameter locking, the result shows that locking the image encoder can achieve the best performance. If the domain of images in the expert data is different than the domain of images in the previous dataset, locking the image encoder seems like a contradiction.

Further, from Table 1, the statistics show the average number of documents and the average number of text/images in documents. But the number of pages and the number of text are too disparate. Take Chevrolet as an example. The car manual has the most pages but less text and images. I am curious about the content in Chevrolet's car manuals.

**Documentation:**

Details are sufficient and code is released. IKEA baselines are provided. But the authors should provide clearly instruction for data usage.

Spelling errors and grammar mistakes:

Line 4: provide extensive an extensive ablation study

Line 13: The Ikea catalogs data was obtained from []

Line 13: The data in consisted of -> The data is consisted of

**Ethics:**

There are no ethical concerns.

**Relation To Prior Work:**

The authors introduce the Vision and Language tasks and Multiple Instance Learning. They also explain and cite several methods of image-text retrieval. However, this paper should cite more datasets about image-text retrieval (e.g. COCO, Conceptual Captions) and compare their difference.

**Summary And Contributions:**

This paper proposes FETA, a retrieval benchmark with a dataset to evaluate Vision and Langauge Pretrained models (e.g. CLIP) on the downstream tasks including car service manuals and IKEA catalogs. Since prior works focus on images of common objects, the authors claim that FETA is the first benchmark aiming at technical documentation with illustrations. They indicate that there is still a large domain gap between previous pre-trained datasets(e.g. ImageNet) and their presented dataset. Also, they extract PDF data based on the DeepSearch tool and propose an automatic annotation of text-image pairs. They evaluate the pre-trained CLIP model on FETA and analyze the experiments by four kinds of settings: zero-shot, one-shot, few-shot, and many-shot. The result shows the effectiveness of Multiple Instance Learning, while there is still room for improvement.

---

> ### Author Response · Authors · 2022-08-18
> **Detailed Response to Reviewer rF7u**
>
> We thank the reviewer for the constructive feedback. Following is our detailed response:
>
> -----
>
> > This paper provides the statistics of the FETA including the number of documents for each car manufacturer and the average text/image numbers in these documents. Nevertheless, each car manufacturer may provide different content in their car manuals. The paper should give a more detailed analysis of the text and images in the documents.
>
> And
>
>
> > Although the authors provide the IKEA-catalogues dataset and provide an experimental evaluation of it, they do not provide any analysis of it.
>
> ### Answer
>
> We thank the reviewer for their feedback. We added more detailed analysis and statistics for both dataset in Section 2 of the revised supplementary material. This includes, among other things, comparison of the proposed datasets to common V&L datasets, i.e COCO, Flickr30K and CC3M.
>
> -----
>
> > The authors indicate that there is a large natural domain gap between their expert domain data and common object data. However, based on their ablation study of parameter locking, the result shows that locking the image encoder can achieve the best performance. If the domain of images in the expert data is different than the domain of images in the previous dataset, locking the image encoder seems like a contradiction.
>
> ### Answer
>
> We thank the reviewer for raising this point, we have revised Sections 4 and 4.4 in the main paper, extending it with intuition on this point as well as enriched Table 4 with additional experiments supporting that intuition. We believe that for the expert tasks we propose for the FETA benchmark it is natural to assume that the huge-scale pre-training data of CLIP (and other FM V&L models) does contain similar style images deep within the long-tail of its data distribution. Hence, useful discriminative features should exist within the pre-trained image encoder model and we need to find ways of uncovering these features to improve CLIP's out-of-the-box performance on the expert tasks. Full fine-tune is of course one way of uncovering those, but, as verified by our experiments (Table 4), it suffers more from automatic supervision noise and overfitting. Locked-image finetune is much safer in these two aspects, indeed attaining better results in some cases, but has less plasticity in the model and as correctly pointed out by the reviewer does not fully exploit the necessary image encoder adaptation. It basically learns to project the output of the image encoder in a better way on the updated text representation space which filters the features only on the output level. However, there are intermediate variants between these two extremes , i.e full FT or locked image. We ran additional experiments using the Low Rank residual Adapters method (LoRA, https://arxiv.org/abs/2106.09685) adopted by us from the NLP LLM domain and applied to V&L models, in this case CLIP. We used LoRA to “interpolate” between full FT and locked image encoder by varying the rank of the added residual adapters from r = 0 to r = 512 equivalent to “locked” and  fully fine-tuned, respectively. Results sampling intermediate values of the rank r are now added to Table 4. As we expected, and corresponding to the above intuition, significantly better results can be obtained for intermediate values of rank r (between 0 and 512).  We thank the reviewer again for suggesting to explore this phenomena further, we believe that these new and interesting findings would constitute an additional contribution to the FETA benchmark, which now includes this additional set of interesting V&L baselines using LoRA adapted to V&L.
>
>
> -----
>
> > After downloading the dataset, the compressed file includes several directories of car manuals without any files in them. The compressed file also includes IKEA images but no texts.
>
> ### Answer
>
> We thank the reviewer for bringing to our attention the missing files in the data. We fixed the issue and re-uploaded the data. Furthermore, we added additional explanations on how to load and examine the data to the supplementary material.

---

> > ### Comment · Reviewer_rF7u · 2022-08-28
> > **Thank you for your response**
> >
> > I appreciate efforts authors made to improve their paper.  The additional experiment in Table 4 is interesting and I appreciate that it was added. The authors also fixed the missing file issue. Further, data quality check shows their annotation pipeline is effective. But I'm curious about human performance. The authors can show the human evaluation of their dataset such as sampling some data and hiring workers to do the multiple choice question. The rebuttal addresses my major concerns, and therefore raise my score to weak accept.

---

> > > ### Author Response · Authors · 2022-08-29
> > > **Human evaluation**
> > >
> > > We thank the reviewer for the feedback and further suggestions.
> > >
> > > We will gladly designs such a user-study to meet the reviewers request. However, due to the upcoming deadline for the discussion, we do not believe that we will have time to publish the results by the end of the discussion period. We will however be able to add this user-study to the camera ready version, if accepted.

---

### Official Review · Reviewer_1gZm · 2022-07-27
**Dataset addresses important gap but lacks sufficient diversity**

**Rating:** 5
**Confidence:** 4
**Correctness:** The claims and approaches seem correct.

**Strengths:**

- The application of large scale pretrained models to technical diagrams is understudied and this work contributes to filling an important gap.
- The authors explore extensive baselines and experimental configurations and demonstrate the significant impact of modeling choices on the resulting performance.
- The paper provides a novel multiple instance learning (MIL) extension to the CLIP model
- The authors release the code pipeline for PDF processing that could enable the generation a broader range of diverse paired image-text datasets


**Weaknesses:**

- While the paper makes claims about tackling the general challenge of benchmark data for expert data domains, the current datasets cover a very narrow set of domains and lack sufficient diversity to make generalizable conclusions about the ability of models to tackle expert data tasks.
- The current framing of the paper tends to overstate the novelty of the application area. While this area is certainly under-studied, there is existing work on non-natural images in technical and expert domains – scientific diagrams, patents, biomedical data, etc. See the “Relation to Prior Work” section below.
- Given the automated nature of the generated labels, further analysis is needed to understand the potential errors and limitations of the dataset. While the comparison of the relative model results between the automated labels and the manual labels provides some confidence in the automated labeling process, manual validation of a sample of annotations to identify common error types and their prevalence would be informative.
- The paper emphasizes the connection to real world practical applications, but the applicability of evaluation tasks to real world use cases is not clear to me. For example, language accompanying illustrations in car manual is unlikely to match the types of queries a real user would use to search for diagrams. A better connection to the potential downstream tasks that such a pretrained model could be used for is needed.
- The observed model generalization performance between manuals from different car manufacturers seems poor, with a substantial amount of training data from the given manufacturer (Many-Shot) required to achieve good performance. This seems to point to a lack of sufficient diversity to enable generalization to new manufacturers which gives me doubts that the results can provide insights that generalize more broadly to other technical diagram applications.


**Additional Feedback:**

- The paper argues that the challenge of the FETA benchmark data is due to the bias towards natural image photos in common training sets rather than sketch-like and synthetic images. If this is the case, what is the intuition for the fact that freezing the image encoder to pretrained CLIP-weights leads to the best model performance? This seems to point to the text representations needing the most adjustment compared with the pretrained models to address this task.
- The paper has a fair amount of typos and proofreading issues (e.g. inconsistent “json” and “JSON”, “pdf” and “PDF”, “resutls" in all of the table captions, etc.)
- There is an inconsistency between main text and supplementary material in terms of the manually annotated dataset: are there 15 or 18 manually annotated documents?
- I understand that space limitations make it difficult to include details on both the car manual and Ikea datasets, but additional comparison of these data sets to understand their commonalities and differences and how they both relate to “expert tasks” would be informative. For example, Ikea catalogues seem more like natural images than the car manuals, which appears to lead to less improvement from the baseline CLIP models relative the improvement seen for the car manuals.


**Clarity:**

The paper is generally structurally well-written although there are some issues with typos and proofreading.

**Documentation:**

The code has been released along with accompanying installation and running instructions. The paper does not include the data checklist after the references. In the version of the data that I was able to download, almost all of the car manual images appear to be missing from the dataset with only one manual directory containing 25 images and the rest of the directories being empty. The Ikea dataset also has some empty directories, with only 9574 total images rather than the 16K listed in the paper.

**Ethics:**

No concerns

**Relation To Prior Work:**

The paper is missing significant prior work in the space of tackling expert tasks in general and those specifically related to sketch-like and synthetic images outside the style of typical foundation model training sets. The authors should include further references in the fields of patent analysis, scientific figures, and biomedical applications. Some potentially relevant citations:
- Kucer, Michal, et al. "DeepPatent: Large scale patent drawing recognition and retrieval." Proceedings of the IEEE/CVF Winter Conference on Applications of Computer Vision. 2022.
- Gryaditskaya, Yulia, et al. "OpenSketch: a richly-annotated dataset of product design sketches." ACM Trans. Graph. 38.6 (2019): 232-1.
- Lee, Jieh-Sheng, and Jieh Hsiang. "Patent claim generation by fine-tuning OpenAI GPT-2." World Patent Information 62 (2020): 101983.
- Pustu-Iren, Kader, Gerrit Bruns, and Ralph Ewerth. "A multimodal approach for semantic patent image retrieval." Proceedings of the 2nd Workshop on Patent Text Mining and Semantic Technologies (PatentSemTech) 2021 co-located with the 44th International ACM SIGIR Conference on Research and Development in Information Retrieval (SIGIR 2021). Aachen, Germany: RWTH Aachen, 2021.
- Jobin, K. V., Ajoy Mondal, and C. V. Jawahar. "DocFigure: A dataset for scientific document figure classification." 2019 International Conference on Document Analysis and Recognition Workshops (ICDARW). Vol. 1. IEEE, 2019.
- Chaudhry, Ritwick, et al. "Leaf-qa: Locate, encode & attend for figure question answering." Proceedings of the IEEE/CVF Winter Conference on Applications of Computer Vision. 2020.
- Mathew, Minesh, et al. "InfographicVQA." Proceedings of the IEEE/CVF Winter Conference on Applications of Computer Vision. 2022.
- Kembhavi, Aniruddha, et al. "Are you smarter than a sixth grader? textbook question answering for multimodal machine comprehension." Proceedings of the IEEE Conference on Computer Vision and Pattern recognition. 2017.


**Summary And Contributions:**

This paper introduces the FETA benchmark for multi-modal (text-to-image and image-to-text) retrieval tasks for technical documentation derived from car manuals and Ikea catalogues and provides a pipeline for the extraction of image-text pairs from other PDF data sources. The authors explore a broad range of baseline modeling approaches, evaluation configurations, and ablations on this task to benchmark the performance on this novel task.

---

> ### Author Response · Authors · 2022-08-18
> **Detailed Response to Reviewer 1gZm (Part1)**
>
> We thank the reviewer for the constructive feedback. Following the reviewer’s comments we propose the following:
>
> -----
>
> > While the paper makes claims about tackling the general challenge of benchmark data for expert data domains, the current datasets cover a very narrow set of domains....
>
> ### Answer
>
> We appreciate the reviewer’s suggestion to increase the variability of the tasks in the  FETA dataset. Following this, we are working on adding new datasets to the FETA benchmark and plan on providing them by the end of the discussion period. We see FETA as a platform which, if accepted, will enable researchers to easily add and use more data from new domains. If accepted, and with the help of the community, we plan on continually expanding the dataset in future versions. Our contribution, in terms of data, is two fold: not only do we present new expert tasks data but we open a convenient gateway to unlimited additional expert data domains arising from various types of documentation, by proposing automatic annotation.
>
> -----
>
> > ... While this area is certainly under-studied, there is existing work on non-natural images in technical and expert domains – scientific diagrams, patents, biomedical data, etc.
>
> ### Answer
>
> We thank the reviewer for bringing these papers to our attention. We have added a discussion paragraph to the end of the main paper’s related work section 5 summarizing the similarities and differences between FETA and these papers. We also shortly summarize our conclusions about each paper below highlighting the differences from FETA. We thank the reviewer for this proposal and hope that with this additional related work discussion the reviewer’s concern was addressed.
>
> 1. Kucer, Michal, et al. "DeepPatent: Large scale patent drawing recognition and retrieval” - This is an interesting paper dealing with  image-to-image retrieval on drawings collected from design patents. Although this is a great example for an expert task for Vision only models, this dataset is not compatible with FETA goals for evaluating V&L FMs as DeepPatent does not contain the accompanying text. We agree that patent data is a great expert data which can be very useful for further analysis of FMs. We are exploring the possibility of adding patent data to future versions of FETA.
>
> 1. Gryaditskaya, Yulia, et al. "OpenSketch: a richly-annotated dataset of product design sketches." - Dataset of line types in design sketches, the paper shows interesting downstream tasks such as deep normal prediction. As opposed to FETA, this dataset is not intended for V&L tasks and does not provide (open) text annotations. Design sketches could be a great addition to future FETA tasks provided that such annotation is collected, e.g. using the automatic annotation technique offered by FETA and an existence of a documentation corpus for design sketches.
>
> 1. Lee, Jieh-Sheng, and Jieh Hsiang. "Patent claim generation by fine-tuning OpenAI GPT-2."  - The authors adapt GPT-2 to generate patent claims. However this work does not deal with images and hence is very different from FETA (that focuses on V&L expert tasks).
>
> 1. Pustu-Iren, Kader, Gerrit Bruns, and Ralph Ewerth. "A multimodal approach for semantic patent image retrieval." -  A very interesting workshop (short) paper also focusing on patent data. Their task is image-to-image retrieval, while leveraging the accompanying text information to facilitate this task. Differently, in FETA we focus on imag-to-text and text-to-image retrieval V&L expert tasks. Unfortunately, neither the code nor the data for this paper were not released which makes it difficult to consider this for FETA, however, we do intend to add patent based expert tasks to the future versions of FETA.
>
> 1. Jobin, K. V., Ajoy Mondal, and C. V. Jawahar. "DocFigure: A dataset for scientific document figure classification." - this paper and accompanying dataset focus on scientific figures and their classification into a fixed set of classes, for example: 3d image, natural image, table, box plot, area chart etc. Differently, FETA focuses on open-vocabulary free-text V&L expert tasks.
>
> 1. Chaudhry, Ritwick, et al. "Leaf-qa: Locate, encode & attend for figure question answering."; Mathew, Minesh, et al. "InfographicVQA”; and Kembhavi, Aniruddha, et al. "Are you smarter than a sixth grader? textbook question answering for multimodal machine comprehension." - are question answering (QA) datasets for reasoning over figures and charts using coupled visual and associated textual data. While VQA is a very interesting task, these works are orthogonal to FETA which focuses on different expert tasks of text-to-image and image-to-text retrieval (which might be a more direct way of evaluating out-of-the-box and finetuned performance of large-scale pre-trained V&L models commonly trained using contrastive objectives). However, we certainly agree that Infographics and Figures are an interesting domain to explore also in the FETA’s context.

---

> ### Author Response · Authors · 2022-08-18
> **Detailed Response to Reviewer 1gZm (Part 2)**
>
>
> -----
>
> > Given the automated nature of the generated labels, further analysis is needed to understand the potential errors and limitations of the dataset. While the comparison of the relative model results between the automated labels and the manual labels provides some confidence in the automated labeling process, manual validation of a sample of annotations to identify common error types and their prevalence would be informative.
>
> ### Answer
>
> We thank the reviewer for their comment regarding automatic annotation error analysis. Following this comment we have performed multiple tests in order to analyze potential errors:
>
> 1. We checked the overlap between the manual annotation and the automatic annotation. We found that in 93%  of the cases the GT manually annotated text is contained in the automatically selected set of up to five boxes. This finding is closely aligned with the Multiple Instance Learning assumption that the MIL bag should contain at least one true positive. We therefore treat the remaining 7% as annotation noise. We added this experiment to section 2.1 of  the revised supplementary material.
>
> 1. We performed a data quality check, presented in Table 1 of the revised supplementary material. In this test we have asked three external reviewers to go over a subset of the automatic annotations. Each reviewer was asked to rate the annotation as good or bad. In 93.6% of the cases at least one reviewer regarded the annotation as good, while in 82.3% of the cases there was a consensus among all reviewers that the annotation is good. For the reviewer's convenience the results are presented here:
>
>   Number of reviewers that marked the automatic annotation as "Good""
>
>   | 1+ | 2+ | 3+ |
>   | ---|---|---|
>   |93.6%|90.2%|82.3%|
>
> 1. We additionally evaluated all the settings of the original experiments (zero-shot, one-shot, few-shot, many-shot) on the manually annotated data and found similar trends to that of the experiments on the automatic annotations. These results are presented in Table 6 of the revised manuscript main paper.
>
>
> -----
>
> >The paper emphasizes the connection to real world practical applications, but the applicability of evaluation tasks to real world use cases is not clear to me. For example, language accompanying illustrations in car manual is unlikely to match the types of queries a real user would use to search for diagrams. A better connection to the potential downstream tasks that such a pretrained model could be used for is needed.
>
> ### Answer
>
> We have additionally tested both MIL models trained on IKEA and car-manuals data with queries collected from real human test subjects. People used phrases such as “change a tire”, “red sofa” and more. We found that for the majority of the queries top-1 or top-2 results were accurate. This implies that models trained on automatic FETA annotations can indeed learn a good multi-modal embedding space which generalizes reasonably well to “human-spoken” text, further underlining its practical value. We have attached a PowerPoint presentation to the supplementary material with examples from our qualitative study.
>
> -----
>
> > The observed model generalization performance between manuals from different car manufacturers seems poor, with a substantial amount of training data from the given manufacturer (Many-Shot) required to achieve good performance. This seems to point to a lack of sufficient diversity to enable generalization to new manufacturers which gives me doubts that the results can provide insights that generalize more broadly to other technical diagram applications.
>
> ### Answer
>
> We thank the reviewer for their insightful comment regarding the generalization capabilities. We would like to point out that this relates to one of the FETA contributions. By proposing an automatic annotation process out of an arbitrary expert task documents corpora containing illustrations, we greatly reduce the cost of annotation for these expert tasks, making it easy to collect the desired data (e.g. for any specific car manufacturer).  We do not claim that solving for one manufacturer should or can be generalized to other technical documents, but rather point to the issue and propose a solution. We added a discussion and clarified these insights in Section 4.3 of the revised manuscript (main paper).
>
> -----
>
> > The Ikea dataset also has some empty directories, with only 9574 total images rather than the 16K listed in the paper.
>
> ### Answer
>
> The Ikea dataset also has some empty directories, with only 9574 total images rather than the 16K listed in the paper. This is corrected in the revised manuscript
>
> -----
>
> > almost all of the car manual images appear to be missing from the dataset with only one manual directory containing 25 images and the rest of the directories being empty.
>
> ### Answer
>
> We thank the reviewer for bringing this important issue to our attention. The download file was fixed and re-uploaded to the same storage location.

---

> ### Author Response · Authors · 2022-08-18
> **Detailed Response to Reviewer  1gZm (Part 3)**
>
>
> -----
>
> > The paper argues that the challenge of the FETA benchmark data is due to the bias towards natural image photos in common training sets rather than sketch-like and synthetic images. If this is the case, what is the intuition for the fact that freezing the image encoder to pretrained CLIP-weights leads to the best model performance? This seems to point to the text representations needing the most adjustment compared with the pretrained models to address this task.
>
> ### Answer
>
> We thank the reviewer for raising this point, we have revised Sections 4 and 4.4 in the main paper, extending it with intuition on this point as well as enriched Table 4 with additional experiments supporting that intuition. We believe that for the expert tasks we propose for the FETA benchmark it is natural to assume that the huge-scale pre-training data of CLIP (and other FM V&L models) does contain similar style images deep within the long-tail of its data distribution. Hence, useful discriminative features should exist within the pre-trained image encoder model and we need to find ways of uncovering these features to improve CLIP's out-of-the-box performance on the expert tasks. Full fine-tune is of course one way of uncovering those, but, as verified by our experiments (Table 4), it suffers more from automatic supervision noise and overfitting. Locked-image finetune is much safer in these two aspects, indeed attaining better results in some cases, but has less plasticity in the model and as correctly pointed out by the reviewer does not fully exploit the necessary image encoder adaptation. It basically learns to project the output of the image encoder in a better way on the updated text representation space which filters the features only on the output level. However, there are intermediate variants between these two extremes , i.e full FT or locked image. We ran additional experiments using the Low Rank residual Adapters method (LoRA, https://arxiv.org/abs/2106.09685) adopted by us from the NLP LLM domain and applied to V&L models, in this case CLIP. We used LoRA to “interpolate” between full FT and locked image encoder by varying the rank of the added residual adapters from r = 0 to r = 512 equivalent to “locked” and  fully fine-tuned, respectively. Results sampling intermediate values of the rank r are now added to Table 4. As we expected, and corresponding to the above intuition, significantly better results can be obtained for intermediate values of rank r (between 0 and 512).  We thank the reviewer again for suggesting to explore this phenomena further, we believe that these new and interesting findings would constitute an additional contribution to the FETA benchmark, which now includes this additional set of interesting V&L baselines using LoRA adapted to V&L.
>
> -----
>
> > There is an inconsistency between main text and supplementary material in terms of the manually annotated dataset: are there 15 or 18 manually annotated documents?
>
> ### Answer
>
> We thank the reviewer for pointing out this typo. There are 15 manually annotated documents. The revised manuscript was edited accordingly.
>
> -----
>
> > :I understand that space limitations make it difficult to include details on both the car manual and Ikea datasets, but additional comparison of these data sets to understand their commonalities and differences and how they both relate to “expert tasks” would be informative. For example, Ikea catalogues seem more like natural images than the car manuals, which appears to lead to less improvement from the baseline CLIP models relative the improvement seen for the car manuals.
>
>
> ### Answer
>
> We thank the reviewer for this insightful comment, we added several comparisons and statistics to Section 2 of the revised supplementary material.
>
>
> -----
>
> > The paper does not include the data checklist after the references.
>
> ### Answer
>
> We thank the reviewer for pointing this out, we added the author checklist to the main paper.

---

### Official Review · Reviewer_QHC7 · 2022-07-28
**Review on FETA**

**Rating:** 7
**Confidence:** 4
**Clarity:** The paper is clearly written.

**Strengths:**

1. The authors claim this work is the first of its kind foundational model for expert tasks dataset and benchmark.
2. This work released an automatic text-image pairs extraction pipeline from product documentation data.
3. Extensive experiments are conducted, and multiple baselines and sufficient ablation tests are provided.
4. Overall, this paper is well structured and statements are clearly presented.

**Weaknesses:**

1. It would be good if there are simple figures illustrating the automatic annotation and the differences between CLIP and those MIL variants of CLIP.
2. The resulting recall score on this documentation dataset is significantly lower than those of normal multimodal datasets - which is good, have you conducted any error analysis? For example, is it because the textual descriptions in the dataset are similar(as all come from a narrow scope) or is it the limitation of the model performance?
3. Typo: pyTorch(P.6 #190)

**Additional Feedback:**

No additional feedback.

**Correctness:**

The paper is constructed in a sound way, and evaluation methods and experiment design for benchmark models are appropriately designed.

**Documentation:**

The paper includes sufficient detail on data collection and ethical discussion.

**Ethics:**

No ethical issues.

**Relation To Prior Work:**

The paper clearly discussed how this work differs from previous works.

**Summary And Contributions:**

This paper proposed the first of its kind foundational models for expert task applications benchmark and dataset focused on evaluating foundational models on expert data tasks. An automatic text-image pairs extraction pipeline from product documentation data is released. Presented a novel Multiple instance learning + contrastive learning approach. Extensive experiments are conducted, including sufficient ablation studies.

---

> ### Author Response · Authors · 2022-08-18
> **Detailed Response to Reviewer QHC7**
>
> We thank the reviewer for the constructive feedback. Following is our detailed response:
>
> -----
>
> > The resulting recall score on this documentation dataset is significantly lower than those of normal multimodal datasets - which is good, have you conducted any error analysis? For example, is it because the textual descriptions in the dataset are similar(as all come from a narrow scope) or is it the limitation of the model performance?
>
> ### Answer
>
> We thank the reviewer for this suggestion. We have added Sections 2.3 and 2.4 to the supplementary material  analyzing the statistics and content of the popular Flickr30k, COCO, and CC3M V&L datasets comparing them to the expert data tasks currently proposed in FETA. From these analysis and the corresponding Tables 2, 3 & 4 in the supplementary, we can clearly see how the content of the expert tasks very significantly differs from the common V&L test sets. While the common datasets are similar in content, in terms of nouns and adjectives concepts, they are very different from the FETA expert tasks in these respects which are in turn also different from one another. We believe the huge-scale pre-training datasets used to pre-train CLIP and other popular V&L models are closer in content to the common datasets, while being significantly different to the terminology, and hence the visual concepts, used for the expert tasks. This might explain the significant decline in recall also noted by the reviewer. We thank the reviewer again for suggesting to include these statistics and analysis, which certainly enriched our paper’s offering.
>
> -----
>
> > It would be good if there are simple figures illustrating the automatic annotation and the differences between CLIP and those MIL variants of CLIP.
>
> ### Answer
>
> We thank the reviewer for this suggestion. We will add more figures illustrating the automatic annotation process and outcomes as well as the CLIP variants by the end of the discussion period.

---

> ### Author Response · Authors · 2022-08-29
> **Additional Figure**
>
> We thank the reviewer for the constructive  and important comments and suggestions.
>
> As promised, we have now added two new figures to the supplementary material illustrating the automatic annotation process and the difference between the CLIP-MIL variants (Supp Figures 1 and 4 respectively).

---

### Official Review · Reviewer_PKHj · 2022-07-29
**Starting Point for Evaluating Foundational Models on Expert Task Applications**

**Rating:** 5
**Confidence:** 3
**Clarity:** This paper is clear and easy to follow.

**Strengths:**

This paper has a clear motivation and potentially would be beneficial for many of the academic community and beyond in the industry. Out-of-box foundational models that can perform practical, real-world expert tasks are of high interest to the industry, and benchmarks that advance the progression in this direction would be an important contribution.

Overall, the paper is easy to follow, and creates a novel expert domain dataset that would be interesting for many.

**Weaknesses:**

There are a few points in data collection methodology and experiment result conclusions that could be further discussed and strengthened:

1. When creating the dataset from raw PDF files, the procedures of data conversion to images and texts are non-trivial and potentially error-prone:
- 1.1 In the supplement, the authors "merged together spatially close text boxes". How is spatially close defined? Given the density of information for data such as car manuals and IKEA catalogs, this could be potentially error-prone.
- 1.2 From sample data, especially for IKEA catalogs, there are many cases where one single large image (e.g. living room or cockpit) fills one or two pages, with multiple text boxes describing parts of the image (e.g. sofa, shelves, speedometer). Would the accuracy be inflated if any texts relevant to this whole image be counted as a correct matching under MIL?
- 1.3 The automatic annotation method only considers contents within the same one page, while in catalogs and manuals, potentially many contents extend beyond one page. It is a fair assumption and minor point - is there any estimate that such assumption would not affect overall data quality?

2. When conducting the experiment, the data is split into five folds, and the results are being averaged and further averaged across manufacturers. Could you explain the choice of using average (e.g. instead of median)?

3. In the baseline method, there are two methods worth discussing: "concatenation" that concatenates all texts in the vicinity of the image into one long text and set it as positive example; and "choose-one" that randomly select one of the texts from the MIL “bag” as the positive example and train using the original CLIP loss. Neither methods are particularly accurate, and it could worsen the results as those texts could belong to other images on the same page. Is it worth it keeping these two methods?

4. Given the complexity and error-proneness of the data retrieval and automatic annotation process, there should be a direct comparison between manual annotation and automatic annotation. Given the same raw dataset, how accurate is the automatic annotation itself? The paper concludes that since the there is similar trend in model performance improvement with automatic and manual annotated data, the automatic annotation method is validated. I believe that there should be more rigor testing the reliability and accuracy of the automatic annotation process.

5. Therefore, when discussing model accuracy, I would be more comfortable with the same raw dataset experiments tested on ground truth or manually annotated data.

6. In the discussion of experiment result, the paper mentions that locked parameter method consistently over-performs, but it is not consistent with the experiment data. For example, in many-shots testing, CLIP-MIL with unlocked parameters perform consistently better in text-to-image task than the locked parameter one.

7. When choosing expert tasks for this paper, the authors note that they largely look for two criteria - abundant text & images, as well as real-world "unnatural" images that would fall under the long tail distribution of the training set for large FMs. A more diverse dataset in different domains might be interesting outside of manufacturing / catalog, such as healthcare. In addition, within the car manual dataset, the diversity of manufacturers could also impact how well the expert tasks are represented. For example, Nissan and Renault are strategic partners that sometimes share the same design - minor details but would be interesting to point out.


**Additional Feedback:**

Minor typos:
- Line 169 "one of" instead of "on of".
- Table 3 and Table 4 description: "results" are misspelled.

In Table 1, one of the metrics is average texts per doc - could you define texts more clearly? Paragraphs?

It is good that authors have considered other FMs since the paper results and benchmark is largely based on CLIP for image-text / text-image retrieval. It might be beneficial to add FLAVA results to the main paper.


**Correctness:**

I have voiced my concerns on the correctness of data collection methodology and experiment result discussion in the previous section above.

**Documentation:**

In the paper supplement, the authors have failed to provide the URL for IKEA catalogs, which seem to be a minor omission. The authors have also mentioned that code and experiment results will be made public once accepted.

**Ethics:**

Since this work mostly builds on public data, I believe it does not introduces new ethical concerns. It might be prudent to contact related manufacturers and IKEA for their permission on data use for research purposes.

**Relation To Prior Work:**

To the best of my knowledge, this work discusses how it differs from previous contributions.

**Summary And Contributions:**

This paper introduces a new benchmark, FETA - Foundational Models for Expert Task Applications, that targets the evaluation and improvements of foundational models on expert domain tasks common in real-world applications. The authors propose an automatic text-image pairs extraction pipeline, evaluation metrics and baseline using Foundational models such as CLIP and FETA-tuned versions of those FMs. The paper concludes that current out-of-box foundational models suffer significant performance drop once given domain-specific expert tasks, and FETA can further advance real-world practical applications of foundational models in expert domains.

Overall, the paper is well motivated and could bring potential real-world impact through improving expert task performance with foundational models. However, some of the discussions regarding methodology and experiment results are not as clear, which leads to my reservation of this paper.

---

> ### Author Response · Authors · 2022-08-18
> **Detailed Response to Reviewer PKHj (Part1)**
>
> We thank the reviewer for the detailed comments and would like to propose answers and additional information.
>
> -----
>
> > How is spatially close defined?
>
> ### Answer
>
> We added section 1.4 to the supplementary material explaining the procedure and implementation details. In short, we increase the length of each bbox edge by a fixed number of pixels (1% of the page length in height and 4x that constant in width), then any boxes which have an overlapping area are merged into a single box which minimally contains both boxes.
>
> ------
>
> > From sample data, especially for IKEA catalogs, there are many cases where one single large image ... Would the accuracy be inflated if any texts relevant to this whole image be counted as a correct matching under MIL?
>
> ### Answer
>
>  We agree with the reviewer's observation that some images in the Ikea dataset contain more than one object, and as such may generally match several texts. We regard the accuracy resulting from comparing to the ground truth generated by the automatic annotation process as an upper-bound for the true, unknown, accuracy that would result if the entire data would be manually annotated and all the compound images are split. We would also like to point out that for the matter of comparing methods, this issue equally helps all methods, making the comparison valid. Furthermore, we provide a manually annotated set to verify the validity of conclusions obtained using the automatic annotation. As demonstrated in our experiments, these conclusions indeed hold for the results obtained by us. Hence, we conjecture it will also be the case for conclusions obtained by future researchers using our automatic data and the pipeline for generating more of such data for new document collections making FETA a useful tool for benchmarking V&L models towards their performance on expert tasks.
>
> -----
>
> > The automatic annotation method only considers contents within the same one page, ... is there any estimate that such assumption would not affect overall data quality?
>
> ### Answer
>
> We thank the reviewer for this comment and would like to make two points:
>
> A. We have added an analysis of the manual annotation overlap with the automatic annotation to section 2.1 of the supplementary material. In the analysis we found that 93% of the manual annotations were indeed covered by the five selected text boxes of the automatic annotation, extracted from the same page. This strengthens our assumption that most of the relevant text does appear on the same page, and the MIL setting is valid.
>
> B. In order to match the remaining 7%, we offer two proposals which we also added to the discussion in Supplementary Sec. 2.1 and will address in future versions of the FETA benchmark, which we will be able to continue to evolve if accepted:
>   * Adding neighboring pages (before and after) to the search space of the automatic annotation
>   * Parsing the documents using titles, section titles, and subsection titles into relevant search areas (which are not limited to specific pages) and using this parsing to match the image-text pairs.
>
> -----
>
> > When conducting the experiment, the data is split into five folds... Could you explain the choice of using average (e.g. instead of median)?
>
> ### Answer
>
> Thank you for this suggestion, we have also evaluated the median metric and present the results in Table 11 of the revised supplementary material. The median results are very similar to the average, rustling in the same conclusions.
>
> -----
>
> > In the baseline method, there are two methods worth discussing: "concatenation" ... and "choose-one" ... Is it worth it keeping these two methods?
>
> ### Answer
>
>  As this is a new dataset, we see the value in providing results also for the straightforward baselines, such as the  “Concat”, “Choose-One” and “Random” CLIP variants. We believe that providing the performance of the straightforward CLIP extensions, besides showing the necessary ablations for the MIL setting, also draws a more complete picture to future researchers using our benchmark in the sense of what works and how.
>
> -----
> >  I would be more comfortable with the same raw dataset experiments tested on ground truth or manually annotated data.
>
> ### Answer
>
> We thank the reviewer for this suggestion of further evaluations on the manually annotated dataset. We ran the requested tests in  the zero-shot, one-shot, few-shot and many-shot settings on the manual data and reported the results in the extended manual annotation evaluation table (Table 3 in the main paper). These new results correspond to similar conclusions, in terms of relative performance obtained on automatically annotated data, and hence strengthen our belief that the experimental results on the automatically annotated dataset are indeed valid for comparing V&L models in terms of their true performance on expert data tasks.

---

> ### Author Response · Authors · 2022-08-18
> **Detailed Response to Reviewer PKHj (Part 2)**
>
>
> >In the discussion of experiment result, the paper mentions that locked parameter method consistently over-performs, but it is not consistent with the experiment data. For example, in many-shots testing, CLIP-MIL with unlocked parameters perform consistently better in text-to-image task than the locked parameter one.
>
> ### Answer
>
> We thank the reviewer for raising this point, we have revised Sections 4 and 4.4 in the main paper, extending it with intuition on this point as well as enriched Table 4 with additional experiments supporting that intuition. We believe that for the expert tasks we propose for the FETA benchmark it is natural to assume that the huge-scale pre-training data of CLIP (and other FM V&L models) does contain similar style images deep within the long-tail of its data distribution. Hence, useful discriminative features should exist within the pre-trained image encoder model and we need to find ways of uncovering these features to improve CLIP's out-of-the-box performance on the expert tasks. Full fine-tune is of course one way of uncovering those, but, as verified by our experiments (Table 4), it suffers more from automatic supervision noise and overfitting. Locked-image finetune is much safer in these two aspects, indeed attaining better results in some cases, but has less plasticity in the model and as correctly pointed out by the reviewer does not fully exploit the necessary image encoder adaptation. It basically learns to project the output of the image encoder in a better way on the updated text representation space which filters the features only on the output level. However, there are intermediate variants between these two extremes , i.e full FT or locked image. We ran additional experiments using the Low Rank residual Adapters method (LoRA, https://arxiv.org/abs/2106.09685) adopted by us from the NLP LLM domain and applied to V&L models, in this case CLIP. We used LoRA to “interpolate” between full FT and locked image encoder by varying the rank of the added residual adapters from r = 0 to r = 512 equivalent to “locked” and  fully fine-tuned, respectively. Results sampling intermediate values of the rank r are now added to Table 4. As we expected, and corresponding to the above intuition, significantly better results can be obtained for intermediate values of rank r (between 0 and 512).  We thank the reviewer again for suggesting to explore this phenomena further, we believe that these new and interesting findings would constitute an additional contribution to the FETA benchmark, which now includes this additional set of interesting V&L baselines using LoRA adapted to V&L.
>
> -----
>
> >... A more diverse dataset in different domains might be interesting outside of manufacturing / catalog, such as healthcare. In addition, within the car manual dataset, the diversity of manufacturers could also impact how well the expert tasks are represented...
>
> ### Answer
>
> We appreciate the reviewer’s suggestion to increase the variability of the tasks in the  FETA dataset. Following this, we are working on adding new datasets to the FETA benchmark and plan on providing them by the end of the discussion period.
>
> -----
>
> The reviewer brought to our attention that we did not provide the URL for the Ikea catalogs. The data was downloaded from https://github.com/ivc-yz/SSR. We added this URL to the manuscript with the correct citation. We thank the reviewer for pointing this out.
>
> -----
>
> The reviewer commented that the FLAVA results should be added to the main paper. We thank the reviewer for this suggestion. The results are now in Table 2 of the main paper.

---

> > ### Comment · Reviewer_PKHj · 2022-08-29
> > **Thanks for the rebuttal on experiment results and conclusions**
> >
> > I appreciate that the authors have fixed one of the most major issues of this paper - reaching the wrong conclusions regarding locking parameters given the experiment data. It would be great if the authors could explain the rationale behind choosing LoRA, and incorporate some of the intuition discussion to the main paper. As many reviewers have mentioned, locking parameters seem contradictory to the core claim of the paper regarding expert domains.
> >
> > In addition, a more detailed discussion of the diversity of the dataset and which domains to be added in the future could also benefit the paper future work section.
> >
> > I also appreciate that authors are able to provide more details on the automatic annotation method and its comparison with manual annotation. However, I still have reservation regarding the general rigor of the method, and in particular the fact that the accuracy would be the inflated upper bound compared to reality even though it is applied across the board.
> >
> > I would raise my score to reflect the improvements the authors made since the initial comments, but based the general experiment setup, dataset diversity, conclusions, novelty and applicability to generic expert domains and the fair amount of minor detail issues, I would hold the paper as a weak reject.

---

> > > ### Author Response · Authors · 2022-08-29
> > > **Further discussion**
> > >
> > > We thank the reviewer for the comments. We believe that the current metric is practical and useful already now as it meets the initial needs of the benchmark for several reasons:
> > >
> > > 1. The automatic annotation process is much more scalable to adding new data and new expert tasks quickly and easily, rather than manual annotation.
> > > 1. Already now, it highlights the shortcomings of the large scale pre-trained FMs to expert data without finetuning.
> > > 1. It allows comparison of various finetune strategies with conclusions transferable to manual metric (as we have shown in Table 3 in the main paper).
> > >
> > > Regarding the intuition behind LoRA, as we discuss in section 4.4 of the revised main manuscript, LoRA allows for a smooth transition (an "interpolation" in the space of optimizations of sorts) between fully fintuning the model and and fully locking the model. The former does not work well most likely due to the limited amount of finetune data (with respect to the huge pre-training data), while the latter works to some extent but is sub-optimal since there is still some domain gap between the training and test data.
> > > Regarding the diversity of the current version of FETA and the discussion on the subject, we point the reviewer to the “Limitations & Future Work.” section at the end of the manuscript where we state that:
> > >
> > > > While the first version of FETA includes close to 150K images and texts, it is still a drop in the ocean of available technical documentation and other documents available for yet unexplored set of different expert V&L data domains….. Future work includes expanding FETA to additional domains and continually evaluating new FMs as they are released to the community.
> > >
> > > We would also like to point out that we plan on adding data from the medical domain, textbook data and others. Furthermore, we are currently working on adding textBook task data leveraging textBookQA. In a preliminary evaluation we have already observed up to 4% I2T gains on top of the pre-trained FMs - will be happy to include the expanded FETA with additional expert tasks in the camera ready version, if accepted.
> > >
> > > We hope that this alleviates the reviewers concerns regarding diversity and correctness of the proposed manuscript and warrants its acceptance.

---

### Author Response · Authors · 2022-08-18
**General Author responses for "FETA: Towards Specializing Foundational Models for Expert Task Applications"**

## General
We thank the reviewers for their productive comments and suggestions. All five reviewers agree that the FETA dataset is the first of its kind and support the motivation behind this work. The reviewers acknowledge the necessity and novelty of such a dataset for comparing foundation models, and the benefit for the academic community (PKHj, QHC7, 1gZm, XJma). The reviewers point out the importance of the automatic dataset annotation process and its potential for scalability (PKHj, QHC7, 1gZm, rF7u, XJma). The extensive baselines and experiments (QHC7, 1gZm, rF7u, XJma) have been noted by the reviewers and specifically the proposed Multiple Instance Learning extension to CLIP has been found novel (QHC7, 1gZm) and effective (rF7u). Finally, the reviewers find the paper clear, well written, and easy to follow.
In the response below, we aim to address all the reviewers’ concerns by adding experiments, datasets, analysis and visualizations.

## Major changes
Following the reviewers' comments we made several major additions to the paper.
For the reviewers' convenience we highlight the added text in color teal both in the revised manuscript and supplementary:
* Added statistics to data.
  * Overlap of automatic and manual annotations (Supplementary Section 2.1).
  * Automatic annotation quality check (Supplementary Section 2.2 Table 1).
  * Comparison of number of tokens and vocabulary size between the datasets (Supplementary Section 2.3 / Table 2).
  * Common Nouns and Adjectives (Supplementary Section 2.4 Table 3).

* Added experiments and clarification (detailed in reviewer responses).
  * LoRA parameter locking ablation. (Table 4 of manuscript).
  * Main results using median instead of average (Supplementary Section 4.7).

* Added discussions and conclusions both in main paper and supplementary.

## Minor changes

* Fixed download link including empty folders issue.
* Fixed typos
* Added author checklist

Next we address the concerns of each reviewer separately.

---

### Author Response · Authors · 2022-08-29
**Additional Modifications and Additions**

We would like to thank the reviewers for the additional feedback and comments. We have further updated the manuscript with additional new results and figures.

We added results for two more V&L methods, ALBEF and VilT, to the Table 2 of the revised  manuscript. We also added the information on these methods to Section 4.3 of the revised supplementary material.

We have added two figures illustrating the automatic annotation flow. (Figure 1 of the supplementary material) and the difference between the CLIP-MIL variants (Figure 4 of the supplementary material).

We would further like to state that, as is written in the manuscript,  we see FETA as an ongoing and ever-expanding benchmark and plan on adding data from the medical domain, textbook data and others. Furthermore, we are currently working on adding textBook task data leveraging textBookQA. In a preliminary evaluation we have already observed up to 4% I2T gains on top of the pre-trained FMs - we will be happy to include the expanded FETA with additional expert tasks in the camera ready version, if accepted.

---

### Meta-Review · Area_Chair_N36M · 2022-09-14

**Recommendation:** Accept
**Confidence:** 5

**Metareview:**

Looking at the rebuttal, the authors have addressed the main concerns of the reviewers. Overall work is novel and beanchmarks in experimental section add a lot of value.

---

### Decision · Program_Chairs · 2022-09-16

Accept